# Combining InSAR and GNSS to Track Magma Transport at Basaltic Volcanoes

**Delphine Smittarello** [1,*] , **Valérie Cayol** [2,3] , **Virginie Pinel** [1] , **Jean-Luc Froger** [2] , **Aline Peltier** [4,5] **and Quentin Dumont** [2]

1   University Grenoble Alpes, University Savoie Mont Blanc, CNRS, IRD, IFSTTAR, ISTerre, 38000 Grenoble, France
2   Laboratoire Magmas et Volcans, Univ. Blaise Pascal, CNRS, IRD, OPGC, 63000 Aubière, France
3   University Jean Monnet, University de Lyon, 42000 Saint-Etienne, France
4   Institut de Physique du globe de Paris, Université de Paris, CNRS, UMR 7154, F-97418 La Plaine des Cafres, France
5   Observatoire Volcanologique du Piton de la Fournaise, Institut de Physique du Globe de Paris, CNRS, UMR 7154, F-97418 La Plaine des Cafres, France
*   Correspondence: smittare@phare.normalesup.org

**Abstract:** The added value of combining InSAR and GNSS data, characterized by good spatial coverage and high temporal resolution, respectively, is evaluated based on a specific event: the propagation of the magma intrusion leading to the 26 May 2016 eruption at Piton de la Fournaise volcano (Reunion Island, France). Surface displacement is a non linear function of the geometry and location of the pressurized source of unrest, so inversions use a random search, based on a neighborhood algorithm, combined with a boundary element modeling method. We first invert InSAR and GNSS data spanning the whole event (propagation phase and eruption) to determine the final geometry of the intrusion. Random search conducted in the inversion results in two best-fit model families with similar data fits. Adding the same time-period GNSS dataset to the inversions does not significantly modify the results. Even when weighting data to provide even contributions, the fit is systematically better for descending than ascending interferograms, which might indicate an eastward flank motion. Then, we invert the GNSS time series in order to derive information on the propagation dynamics, validating our approach using a SAR image acquired during the propagation phase. We show that the GNSS time series can only be used to correctly track the magma propagation when the final intrusion geometry derived from InSAR and GNSS measurements is used as an a priori. A new method to extract part of a mesh, based on the representation of meshes as graphs, better explains the data and better accounts for the opening of the eruptive fissure than a method based on the projection of a circular pressure sources. Finally, we demonstrate that the temporal inversion of GNSS data strongly favors one family of models over an other for the final intrusion, removing the ambiguity inherent in the inversion of InSAR data.

**Keywords:** InSAR; GNSS; Piton de la Fournaise; dike propagation; inversion; joint inversion

## 1. Introduction

The advent of Synthetic Aperture Radar Interferometry (InSAR) in the mid-1990s shed a new light on magma plumbing systems, which connect deep magma reservoirs to the surface at volcanoes [1–3]. Due to its exceptional spatial coverage, this remote sensing geodetic method can be used to produce ground surface displacements maps, which has led to a significant improvement in the imaging of magma storage zones. In particular, we now have a better characterization of the specific reservoir

shapes (e.g., Reference [4]), the possibility to distinguish several active storage zones beneath active volcanoes (e.g., Reference [5]), as well as statistical studies on magma reservoir depth [2] and relative location compared to the volcanic edifice [6]. This has also led to better estimates of associated volume changes. In addition to the characterization of the storage zone geometry, another advance has been developing imaging of magma transport features. Magma can accumulate for months to years in a storage zone [7]. However, once the pressure inside this reservoir reaches a critical threshold, rupture is induced and magma starts to propagate towards the surface. It flows inside planar intrusions formed by fracturing of crustal rocks, called either dikes or sills depending on their orientation with respect to pre-existing features. This ascent is usually much quicker than the storage phase and lasts between a few tens of minutes to a few days [8–10]. Once the eruption has ended, part of the magma remains trapped and cools at depth leading to ground displacements detectable by InSAR. Co-eruptive interferograms, calculated by combining images acquired before the transport phase onset and after the eruption, contribute to our knowledge of magmatic intrusion geometries, revealing complex shapes which are probably influenced by the local stress field [11,12].

The still relatively poor temporal resolution of Synthetic Aperture Radar (SAR) data remains a clear limitation for their use in tracking magma propagation. This phase, which is crucial to understanding the dynamics of magma emplacement, is usually recorded by continuous, ground-based sensors such as GNSS stations, tiltmeters or seismometers for example, References [9,13–15]. It is not always possible to place ground-based captors in ideal locations with respect to the intrusion due to field conditions [16]. Moreover, seismicity does not always permit to track magma propagation, specifically when microearthquakes are too shallow and have a too small magnitude [17]. However, SAR temporal resolution has improved appreciably: the usual revisit time was reduced from one month to 6 days with the launch of the Sentinel–1 satellite constellation. There are a few cases for which SAR images have been acquired during a propagation phase and they all yield exceptional results. One early example is the sill emplacement preceding the 1999 Eyjafjallajökull eruption. The SAR acquisition was favored by a long-lived propagation phase probably linked to limited magma driving pressure [10]. Similarly, a SAR image acquired during the 2009 eruption at Fernandina volcano, Galapagos [18] provided insight into the complex transition from circumferential intrusions to radial eruptive fissures. Also the May 2016 eruption at Piton de la Fournaise volcano was recorded by an intermediate Sentinel-1 acquisition, which occurred only 2 h before the onset of the eruption.

With the increase in the number of radar imaging satellites, InSAR data are available with different Lines-of-Sight (LOS) and different precisions depending on the radar wavelength [19]. When crustal processes occur suddenly, it is possible to use all the interferograms that captured the event. However, for slow processes such as fault creep, viscous relaxation and shallow magma reservoir inflation [20,21], or when several processes take place successively [22], this approach may lead to unreliable results or conceal subtle processes. Here, we compare two strategies, one where we invert interferograms that most closely capture an event and one that uses interferograms covering a specific time period.

Inverse models are used to analyze displacements, combining forward models and inversions. Inversion of surface displacements is inherently non unique. For instance, Dieterich and Decker [23] showed that, when sources are axi-symmetrical, the source shape and depth can not be determined using vertical displacements alone but that both the vertical and horizontal displacements are needed to accurately determine these characteristics. Here, the problematic is different because the sources are non axi-symmetrical and 3D displacements can be computed. Since, this study is of a fissure eruption, the source of unrest is a fracture, which is a non axi-symmetrical source. Non uniqueness in the surface response may be caused by material properties [24,25] but for a given material response, there is a unique surface displacement for each individual fracture. Moreover InSAR provides displacements along several ascending and descending LOS and GNSS gives 3D displacements, thus the combined use of these two types of data can be used to describe 3D displacements [26]. Non unique source determination comes from inverse modeling, which inherently accounts for errors

in data and models. The main sources of error in InSAR and GNSS data derive from atmospheric conditions (water content of the troposphere and ionosphere) and models are simplifications of a complex reality: the number of sources is unknown and their shapes are probably more complex than assumed. A previous study of the 26 May 2016 eruption at Piton de la Fournaise [16] showed that, despite the very dense network of permanent GNSS stations present on this volcano, magma propagation could only be successfully tracked when adding a priori information on the intrusion shape provided by the inversion of co-eruptive interferograms. This conclusion was reached using the intermediate Sentinel-1 acquisition, which allowed different strategies to invert GNSS data to be quantitatively evaluated. Here, we extend our previous study on the combined use of InSAR and GNSS data and we investigate the usefulness of GNSS time series in resolving the non-unique inverse models obtained from InSAR data.

Time series of displacements, whether provided by continuous GNSS and tilt measurements [27] or by InSAR [28], are usually analyzed using kinematic inversions, where amplitudes of dislocation sources [29] are determined from minimizing the sum of the L2 norm of the weighted residual on displacements and spatially smoothed dislocation amplitudes. This requires as many parameters as there are dislocations, which leads to statistically less likely solutions [30] and solutions physically less meaningful than pressure sources [31]. In our previous study [16], we addressed these issues by determining the pressurized area corresponding to the active intrusion, where the a priori intrusion was derived from InSAR and a circular pressure source was projected onto the mesh representing this intrusion. This method had two drawbacks: 4 parameters were needed when the number of permanent GNSS measurements is 30 and a projection from a plane on the intrusion geometry was needed, which could exclude part of the intrusion and thus induce errors if the intrusion geometry was curved.

In this study, we focus on the methodological aspects of jointly inverting InSAR and GNSS (campaign and time series) data covering the May 2016 Piton de la Fournaise eruption in order to track magma propagation. Because of the non-unique nature of inversions, data covering the whole eruption leads to two families of equally well fitting intrusion models. The influence of the data weightings on the inverted intrusion is also studied. We next use times series of GNSS data to determine the dynamics of the magma propagation for these families of models and to investigate if time series help in the determination of the most likely family of models. Finally, three methods are compared to track magma in the intrusion, one with no a priori and two using the a priori geometry determined from data covering the whole eruption. One of these methods is a new strategy, which uses a graph-based representation of meshes to define the pressurized part of the intrusion. Results from this new method are compared with previous results.

## 2. The 26 May 2016 Eruption of Piton De La Fournaise Volcano

Piton de la Fournaise is a highly active basaltic volcano located on the eastern part of Reunion Island (France), in the Indian Ocean (Figure 1). The summit cone, at an elevation of 2631 m, lies in an uninhabited caldera-shaped depression called the Enclos Fouqué caldera. InSAR revealed a seaward motion (1.4 m) of the eastern flank [32] of this volcano during the major April 2007 eruption [33,34]. GNSS stations installed in this area after the eruption, combined with InSAR monitoring, indicate an ongoing eastward motion of about $\sim$2 cm·year$^{-1}$ [35]. A new eruption cycle started in June 2014 [36,37]. Since then 16 eruptions have occurred on the flanks of the cone, inside the caldera-shaped depression including the May 2016 eruption studied here. After 10 days of minor inflation and a low seismicity rate, a seismic crisis started on May 25 at 19:40 (all times are U.T.C) accompanied by rapid deformation, which corresponded to magma transfer towards the surface [16]. At 04:05 on May 26, a fissure opened, 2800 m to the southeast of the summit cone, at an elevation of 1890 m producing lava fountaining and a 0.5 Mm$^3$ lava flow, which is an unusually small volume for a Piton de la Fournaise eruption [8]. The eruption stopped after 27 h. InSAR acquisitions and GNSS measurements provide an extensive dataset which can be used to understand the magma transfer during this eruption [16].

Seismic data are also available. However, in this case earthquakes being too small and shallow, precise relocation does not allow to track magma at the depth indicated by InSAR and GNSS data [16,17].

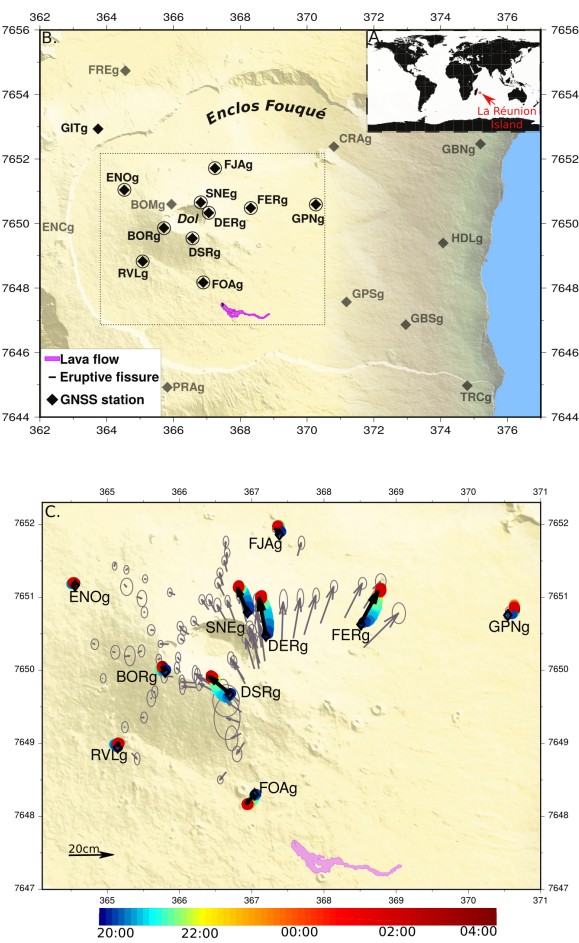

**Figure 1.** Piton de la Fournaise (PdF) setting. (**A**) Location of Reunion Island; (**B**) Shaded topography of PdF volcano and location of the GNSS monitoring network. Black and grey diamonds represent the locations of the permanent GNSS stations. The black circled diamonds refer to the 10 GNSS stations used in this study. The GITg station (black diamond to the northwest) is the reference used here for the GNSS campaign and for the epoch-by-epoch processing. The May 2016 lava flow is shown in purple. 'Dol' marks the Dolomieu crater; (**C**) Zoom corresponding to the area indicated by a black box in B. Black arrows represent the cumulated horizontal displacement of the 10 GNSS stations between 25 May at 17:00 and 26 May at 04:05. Standard deviations for horizontal and vertical components are 1–2 cm and 3-4 cm, respectively. Associated colored circles represent the temporal evolution of the horizontal displacement for the GNSS stations. Grey arrows represent displacements from the GNSS campaign network between 31 August 2015 and 27 May 2016, with ellipses for the 95% confidence intervals. Coordinates are in kilometers (WGS84, UTM 40S). Modified after Reference Smittarello et al. [16]

## 3. Geodetic Measurement Description

### 3.1. InSAR Data

Synthetic Aperture Radar (SAR) images are systematically acquired by various satellites (Sentinel-1 in Strimap Mode, Cosmo Skymed, ALOS2, TerraSar-X) [38] within the framework of the national monitoring service, OI$^2$. For the 26 May 2016 event, images acquired by both Sentinel-1 and Cosmo-Skymed are available.

All interferograms were produced using the DIAPASON software [39], removing the orbital contribution based on the precise orbit state vectors provided by Space Agencies as well as the topographic contribution using a 5 m Lidar DEM, produced by the French Geographic Institute (IGN) in 2008–2009. The interferograms were unwrapped using the Snaphu algorithm [40]. They were then detrended and shifted such that a null displacement in the Enclos Fouqué caldera (see location on Figure 1) was used as a reference. Before being used as inputs for the inversion, the interferograms were subsampled using a quadtree decomposition algorithm [41,42]. The number of points in each region was proportional to the displacement amplitude and to the displacement standard deviation in the region. Thresholds for the amplitude and the standard deviation were adjusted to keep a balanced number of points of around 500 per interferogram.

Cosmo SkyMed acquisitions on 29 February and 31 May, enabled two interferograms to be created (CSKA and CSKD, Figures 2 and 3) spanning 3 months which covered the inter-eruptive period, the unrest, the eruption and 5 days after the eruption. Six Sentinel–1 acquisitions on 19–20 April, 25–26 May and 6–7 June, were used to compute 6 interferograms.

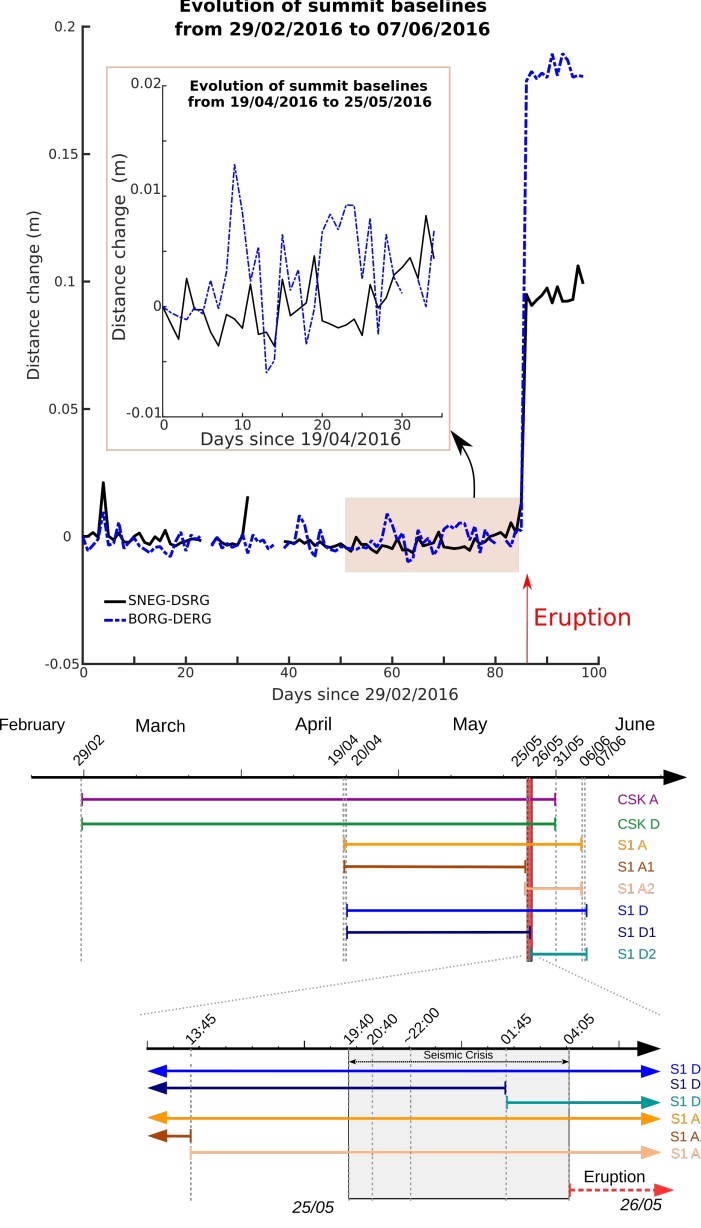

**Figure 2.** (**Top**) Temporal evolution of baseline changes recorded between two pairs of permanent GNSS stations (SNEG and DSRG and BORG and DERG; see Figure 1 for station locations) between 29 February

2016 (date of the first SAR acquisition used in this study) and 07/06/2016 (date of the last SAR acquisition used in this study). Distance samples are daily averages. Inset corresponds to a zoom on the pre-eruptive baseline changes spanned by Sentinel–1 data. The eruption onset is shown by a red arrow; (**Middle**) Coloured bars represent the time spanned by each of the 8 interferograms used in this study. Modified after Smittarello et al. [16]. The vertical red area represents the eruptive crisis; (**Bottom**) Zoom. The vertical dashed line at 01:45 marks the time of the intermediate SAR acquisition by Sentinel–1. CSK = Cosmo-Skymed; S1 D1 = Sentinel–1 descending InSAR data spanning the beginining of the eruptive crisis; S1 D2 = Sentinel–1 descending InSAR data spanning the end part of the eruptive crisis; S1 A1 = Sentinel–1 ascending InSAR data spanning the pre-eruptive period; S1 A2 = Sentinel–1 ascending InSAR data spanning only the eruption.

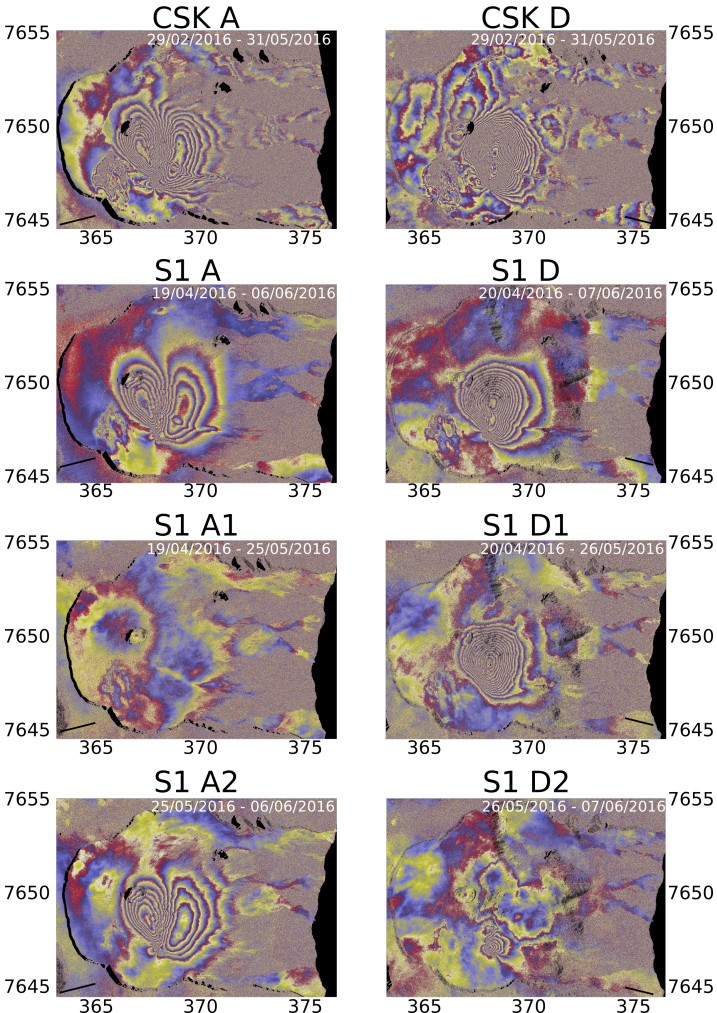

**Figure 3.** Wrapped interferograms used in this study. (**Left column**) Data acquired on ascending orbits; (**Right column**) Data acquired on descending orbits; (**First row**) Cosmo-Skymed data (X-band) spanning the whole eruptive period; (**Second to Fourth rows**) Sentinel-1 InSAR ascending and descending data (C-band); (**Second row**) Sentinel-1 data spanning the whole eruptive period; (**Third row**) Sentinel-1 data spanning the first time period; (**Fourth row**) Sentinel-1 data spanning the second time period; For Cosmo-Skymed data, fringes represent 1.5 cm LOS displacement, while for Sentinel-1 data, fringes represent 2.8 cm LOS displacement. S1 D1 = Sentinel-1 descending InSAR data spanning the beginining of the eruptive crisis; S1 D2 = Sentinel-1 descending InSAR data spanning the end part of the eruptive crisis; S1 A1 = Sentinel-1 ascending InSAR data spanning the pre-eruptive period; S1 A2 = Sentinel-1 ascending InSAR data spanning only the eruption.

In ascending LOS, the Sentinel-1 May 25 acquisition occurred just before the beginning of the crisis (Figures 2 and 3), enabling two interferograms to be computed. Interferogram S1 A1 only records the unrest period while interferogram S1 A2 spans the eruption and a few days after. A third ascending interferogram, S1 A, spans the whole period between 19 April and 6 June recording the unrest, the eruption and a few days after. In descending LOS, the Sentinel-1 26 May acquisition occurred during the crisis (Figures 2 and 3), allowing interferogram S1 D1 to be computed, which spans the pre-eruptive period and the beginning of the crisis and interferogram S1 D2, which covers the end of the crisis and the following few days. A third descending interferogram, S1 D, spans the whole eruptive period.

*3.2. GNSS Data*

Surface displacements of Piton de la Fournaise volcano are continuously monitored by a permanent GNSS network consisting of 24 stations, operated by the Piton de la Fournaise Volcanological Observatory (OVPF). In this study, we used only the 10 stations located in the summit part of the Enclos Fouqué area, which were close enough to the intrusion to capture the unrest associated with the eruption (see Figure 1).

Processing was carried out with the GAMIT/GLOBK software package [43] to determine a daily position for each station. The station REUN, located 17 km away from the volcano was used as a reference to correct for Earth plate motion. Time evolution of two GNSS baselines located at the summit is shown on Figure 2. No significant deformation was recorded during the period spanned by InSAR data prior to and after 25–26 May 2016. Only a very subtle inflation signal (less than 1 cm) can be inferred during the 10 days preceding the eruption (see top part of Figure 2).

In order to analyze surface displacements due to magma propagation during the crisis, an epoch-by-epoch processing in differential mode was also performed with the TRACK software. The station GITg on the upper edge of the Enclos Fouqué caldera was used as a reference. The result is a time series sampled every 30 s. Because of the limited elevation difference between the stations, we assumed that there were no differences in the atmospheric delay for the area (6 km × 4 km). The automatic tropospheric correction was removed during the processing. Then, to correct sidereal effects, GNSS measurements from the 15 days preceding the crisis were cut into slices spanning 23 h 56 min 4 s (the revolution period for GNSS satellites) and stacked. The average signal resulting from multipath reflections on the antenna was removed from the time series. At last, a principal component analysis (PCA) was also conducted to remove correlated noise between stations. Smoothing was applied using a 30 minutes window filtering. More details can be found in Smittarello et al. [16].

A network of 80 points was also measured after each eruption during field campaigns (Figure 1). It provides a dense map of the 3D-displacements that occurred between 28 August 2015 and 30 May 2016. This data was included in the inversion to better constrain the north-south component of the surface displacements which is not always easy to retrieve with InSAR data alone [38,44].

## 4. Inverse Models

Ground displacements measured by InSAR and GNSS were modeled using a 3-D Mixed Boundary Element forward method [45,46] which assumes that the medium is linear, elastic and homogeneous. Fast and precise computations are produced by the combination of a direct method and a displacement discontinuity method used for the ground surface and fractures, respectively. The model can take tensile cracks and shear fractures under a realistic topography into account [47–49]. Boundaries were meshed using triangular elements. To avoid edge effects, the ground surface mesh was five times larger than the deformed area. To provide the best compromise between computation time and accuracy, the most deformed boundaries were finely meshed, while away from the deformation zone, coarser meshes were used. A Young's Modulus of 5 GPa and a Poisson's ratio of 0.25 were assumed for the Piton de la Fournaise edifice, following Cayol and Cornet [46], Fukushima et al. [48].

To take advantage of the whole data set acquired during this eruption, we developed a strategy for inversion that consisted of several steps (Figure 4). First, we performed a global inversion that is, an inversion of the cumulated InSAR and GNSS displacements recorded between two dates: one before the eruptive crisis onset, that is, before the onset of the magma propagation and one after the eruption had ended. In this study, we refer to this global inversion as the "static inversion" as it is intended to reproduce the final geometry of the magma that was trapped at depth after the eruption had resumed. Second, we performed a temporal inversion, that is, inversions of the cumulated displacements for 29 time intervals between 20:40 on 25 May and 4:00 on 26 May, from GNSS time series. This temporal inversion was used to track the dynamics of the magma propagation before it reached the surface and the eruption began. The last step was a validation of the methods used for the temporal inversion and of the model family used as an a priori. It compared the misfits and models inverted at a date for which an intermediate InSAR acquisition was available, allowing the displacement of the first part of the eruption to be measured.

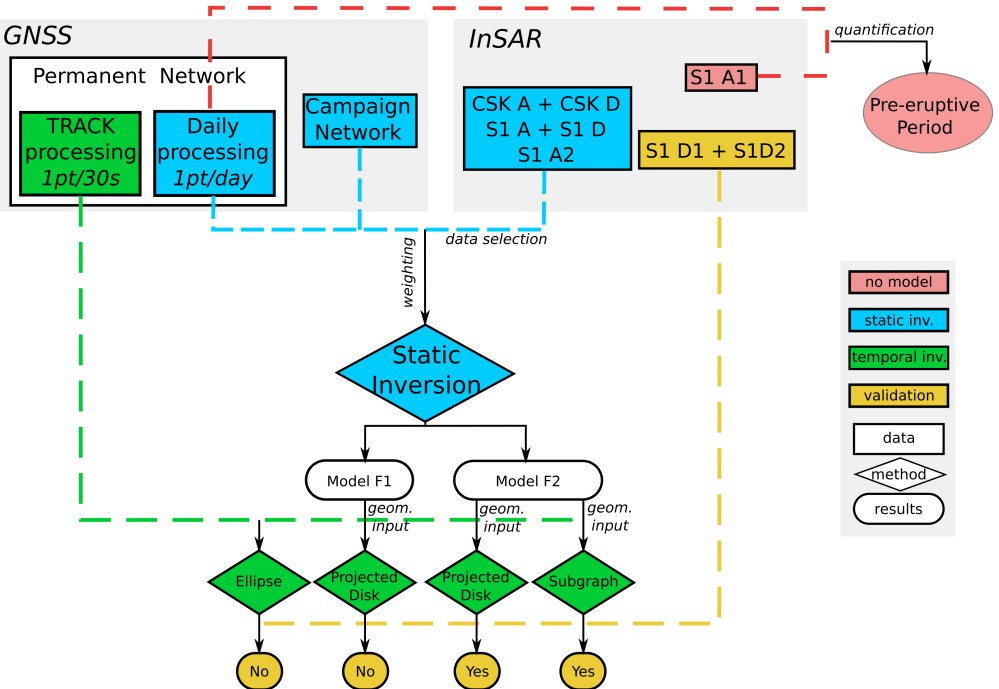

**Figure 4.** Use of the different data sets in our modeling strategy.

*4.1. Inversion and Data Weighting Through the Covariance Matrix*

Inversion of ground displacements to determine the best-fitting deformation model requires minimization of the difference between the observations and the model results. This difference is called a misfit and we calculated it in a least square sense (L2-norm) as :

$$\chi^2 = (\mathbf{d_0} - \mathbf{G(m)})^T \mathbf{C_d}^{-1} (\mathbf{d_0} - \mathbf{G(m)}) \tag{1}$$

where $\mathbf{d_o}$ represents the vector of observed displacements, $\mathbf{G(m)}$ is the vector of modeled displacements with $\mathbf{m}$ the set of model parameters and $\mathbf{G}$ represents the model prediction. $\mathbf{C_d}$ is the covariance matrix for the data, which accounts for correlated noise related to the atmospheric contribution to InSAR data and for the variance of GNSS data.

The atmospheric contribution results in correlated random noise, which can be characterized by the autocorrelation function or covariance function [48,50], expressed as:

$$C(r) = \sigma_d^2 exp(-r/a) \tag{2}$$

where $r$ is the distance between two points, $\sigma_d^2$ is the noise variance and $a$ is the correlation length. Following the study of Fukushima et al. [51], the data covariance matrix $\mathbf{C_d^{SAR}}$ can be computed from this function using a noise variance $\sigma_d^2 = 5 \times 10^{-4} \text{ m}^2$ and a correlation distance $a = 850 \text{ m}$. This results in a diagonal dominant matrix where diagonal terms correspond to $\sigma_d^2$. In addition to InSAR data, GNSS measurements were used from both permanent and campaign networks. We assumed there was no correlation between the GNSS stations, so that the covariance matrix $\mathbf{C_d^{GNSS}}$ was diagonal and was a function of the variance.

$$\mathbf{C_d^{GNSS}} = diag\left(\sigma_d^2\right) \tag{3}$$

Typical magnitude of the GNSS variance was $\sigma_d^2 = 2 \times 10^{-4} \text{ m}^2$ for horizontal components and $\sigma_d^2 = 9 \times 10^{-4} \text{ m}^2$ for vertical components.

The data vector $\mathbf{d_o}$ and covariance matrix on data $\mathbf{C_d}$ combining InSAR and GNSS data are given, respectively, as :

$$\mathbf{d_o} = \begin{pmatrix} \mathbf{d_o}^{GNSS} \\ \mathbf{d_o}^{SAR} \end{pmatrix} \tag{4}$$

$$\mathbf{C_d} = \begin{bmatrix} \mathbf{C_d^{GNSS}} & 0 \\ 0 & \mathbf{C_d^{SAR}} \end{bmatrix} \tag{5}$$

Inversions consist of two stages : a search and an appraisal stage. In the search stage, the misfit is minimized using a near-neighborhood inversion algorithm [52]. In the appraisal stage, model uncertainties are estimated by resampling the population of models obtained in the search stage following the Bayesian inference [53].

Because the misfit function is a L2-norm, it favors models that explain the largest displacements. The matrix covariance characterizing the data is used to account for errors and intrinsic noise affecting the data. A proper definition of the covariance matrix should weight data according to their own errors in the inversion process. However, InSAR data are projections of a single displacement field along the LOS direction. Due to this projection, a displacement vector will produce LOS displacement of different amplitudes on ascending and descending LOS. Since a null LOS displacement can mean either no displacement or displacement orthogonal to the LOS, there is no reason to give a lower weighting to datum of lower amplitude. To give equal weighting to each dataset and avoid overweighting one LOS over the other or the GNSS data when they were added, we chose to conduct inversions in which misfits of the different datasets were scaled by their reference values. We defined the reference value as the value computed from a null model $\chi_{ref}^2$ as

$$\chi_{ref}^2 = \mathbf{d_o}^{\mathbf{T}} \mathbf{C_d}^{-1} \mathbf{d_o} \tag{6}$$

This defined the misfit relative to this reference value $\chi^2(\%)$ as

$$\chi^2(\%) = \frac{\chi^2}{\chi_{ref}^2} \times 100 \tag{7}$$

This value being independent of the total amplitude of displacements, it is useful for comparing inversion results at different time steps of the temporal inversion.

We conducted inversions according to a weighted misfit scaling each dataset by its own reference misfit:

$$\chi_w^2 = \left( \sum_{i=1}^{N} \frac{\chi^{2^i}}{\chi_{ref}^{2^i}} + \frac{\chi^{2^{GNSS}}}{\chi_{ref}^{2^{GNSS}}} \right) \times \frac{\chi_{ref}^2}{N+1} \tag{8}$$

where $i$ refers to the $i^{th}$ interferogram and $N$ is the total number of interferograms. This scaling is equivalent to applying different scaling factors of $\chi_{ref}^{2^i}/\chi_{ref}^2$ with $i = 1, .., N$ and $\chi_{ref}^{2^{GNSS}}/\chi_{ref}^2$ to the

covariance matrices of the InSAR and the GNSS data, respectively. In terms of Bayesian inference, this scaling gives the same likelihood to each of the InSAR and GNSS datasets and each of these datasets has the same contribution to the posterior probability density distribution.

*4.2. Static Inversion*

Given the location of the eruptive fissure, the geometry of the displacement source can be described by nine parameters, which define a quadrangle-shaped intrusion, linked to the surface by a single fissure. This quadrangle can be curved in the along strike and dip directions [16,51].

To determine the static geometry of the magma intrusion which fed the 26 May 2016 eruption, we used ground deformation data from interferograms along four different LOS, daily solutions of the permanent GNSS network and the GNSS campaign network. Of the eight available interferograms, five span the total eruption period (CSKA and CSKD, S1 D, S1 A and S1 A2, on Figure 3). S1 A and S1 A2 are taken along the same LOS but the pre-eruptive period spanned by S1 A is longer. We inverted S1 D, CSK D and CSK A but had two different options for the fourth interferogram in the ascending direction: (1) using the data closest to the eruption time (S1 A2) and (2) using the interferogram (S1 A) most consistent with the acquisition dates of the three other interferograms. We chose to invert the interferograms directly rather than use the 3D components reconstruction of the displacement field in order to reduce the errors induced from using several interferograms spanning different time periods. We decided not to use the whole dataset because the use of different data in the same LOS could have led to over weighting of one LOS.

In total we run seven inversions (Table 1). With inversions 01, 02a and 02b, we checked whether the inflation during the 10 days of unrest could be neglected by comparing the influence of interferograms S1 A and S1 A2 on the inversion results (Table 1). In inversion 04a, 04b and 05, we tested whether the addition of GNSS data, which adds constraints to the north-south displacement, led to a different, better constrained source. Inversions 02a and 02b (and 04a and 04b) were conducted with the same initial conditions in terms of data used, weighting and parametrization, in order to test the variability of the inversion due to the random search. Lastly, in inversions 03, 04a and 04b, we tested the weighting method to investigate if different inverted sources were obtained and to gain insight into the origin of discrepancies between independent datasets.

**Table 1.** Covariance weigthing (*Cd* is defined based on Equations (4) and (5), 'No' means that no specific additional weighting is applied whereas $\chi^2_{ref}$ means that the misfit is scaled by the misfit of the null model), geometry of the best model (for families definition see Section 5.1.1) and % of Explained data (see Equation (9)) computed for each interferogram spanning the eruption. Numbers in bold are for data used in the inversion, values in small italics are computed a posteriori for data not used in the inversion. Inv01 was computed with InSAR data S1 A2, which does not cover the pre-eruptive period. Inv. 01 to 03 were computed without including the GNSS data. Inv02a* is the preferred model presented in Smittarello et al. [16].

| Model | Covariance Weighting | Family | GNSS | InSAR | | | | | |
|---|---|---|---|---|---|---|---|---|---|
| | | | | S1D | S1 A | S1 A2 | CSKD | CSKA | Total |
| **Inv01** | No | F1 | *78* | **97** | *81* | **75** | **97** | **83** | 93.5 |
| **Inv02a*** | No | F2 | *73* | **95** | **82** | *72* | **95** | **82** | 92.5 |
| **Inv02b** | No | F2 | *75* | **96** | **83** | *76* | **95** | **83** | 93.0 |
| **Inv03** | $\chi^2_{ref}$ | F2 | *73* | **96** | **86** | *77* | **96** | **87** | 94.1 |
| **Inv04a** | $\chi^2_{ref}$ | F1 | **77** | **95** | **82** | *74* | **95** | **87** | 93.5 |
| **Inv04b** | $\chi^2_{ref}$ | F2 | **71** | **96** | **83** | *72* | **96** | **85** | 93.5 |
| **Inv05** | No | F1 | **83** | **94** | **67** | *57* | **95** | **75** | 90.3 |

*4.3. Temporal Inversion*

In order to model the dynamics of the magma intrusion as it propagated to the surface, GNSS data from the permanent network were inverted with the same boundary element program and the same inversion algorithm as for the static inversion. We chose to invert the whole time series running 29 independent inversions computing the cumulated displacement for progressively longer time periods as the displacement rate decreased. The start time was 17:00 and the end time varied from 20:40 on 25 May to 4:00 on 26 May. Because of the limited number of GNSS measurements (30 including the east, north and vertical directions), the parametrization of the source had to have less parameters than for the static inversions. Moreover, contrary to the static inversion, the source description could not rely on the eruptive fissure trace at the surface as the magma had not yet reached the surface. We tested three different methods to define the source. The Ellipse and the Projected Disk methods have been presented in greater detail in Smittarello et al. [16] while the Subgraph method is new. The Ellipse method does not use the a priori gained from the static inversion, while the Projected Disk and the Subgraph method relies on it.

4.3.1. A Method without Any Geometric a Priori from the Static Inversion: The Ellipse Method

The so-called Ellipse method uses a planar elliptical source shape with eight parameters to define its geometry: location of the center (x,y,z), orientation ($\phi$, $\theta$, azimuth) and size (2 axes lengths) and one parameter for the overpressure $P$. We used this method to check whether models based solely on GNSS data are reliable, given that the permanent GNSS network on Piton de la Fournaise is one of the densest in the world for a volcano and that often, continuous GNSS data are inverted omitting the information that could be brought by available InSAR data [54,55].

4.3.2. A First Method with a Geometric a Priori: The Projected Disk Method

The Projected Disk method relies on the mesh resulting from the static inversion. We assumed that the displacement recorded by GNSS at each time step could be explained by the pressurization of a subpart of the final mesh. This a priori assumes that the tail of the dike did not close up after the magma had intruded, which seems a reasonable assumption considering that magma is viscous, quickly solidifying and that lateral intrusions were probably emplaced above the level of neutral buoyancy implying that they remained under pressure [51,56]. An average plane was defined from this original mesh. We inverted a circular area onto this plane, for a given location ($x,y$), radius ($R$) and overpressure $P$. The projection of this circle onto the original mesh was chosen to select a subpart of the original mesh, which defines the new pressure source. This method works well if the static geometry is simple (i.e., if it is possible to define a mean plane and a projection). However, with a more complex geometry, in particular if there are strong curvatures of the fracture, the projection may lead to an absurd geometry.

4.3.3. A Second Method with a Geometric a Priori: The Subgraph Method

In order to address the projection issue related to the Projected Disk method above, we designed a new method. Here, a mesh is considered as a graph that is, a set of nodes linked by edges (Figure 5). For instance, for the graph on the left-hand plot which is constituted of 7 nodes, the list of edges is $[(1,2),(1,3),(1,4),(4,5),(4,6),(6,7)]$. A matrix representing the distances between nodes in the graph can then be constructed from this list. The distance is expressed as the minimum number of edges required to draw a path between two nodes. For the left-hand plot (Figure 5), the matrix of distance has the following values.

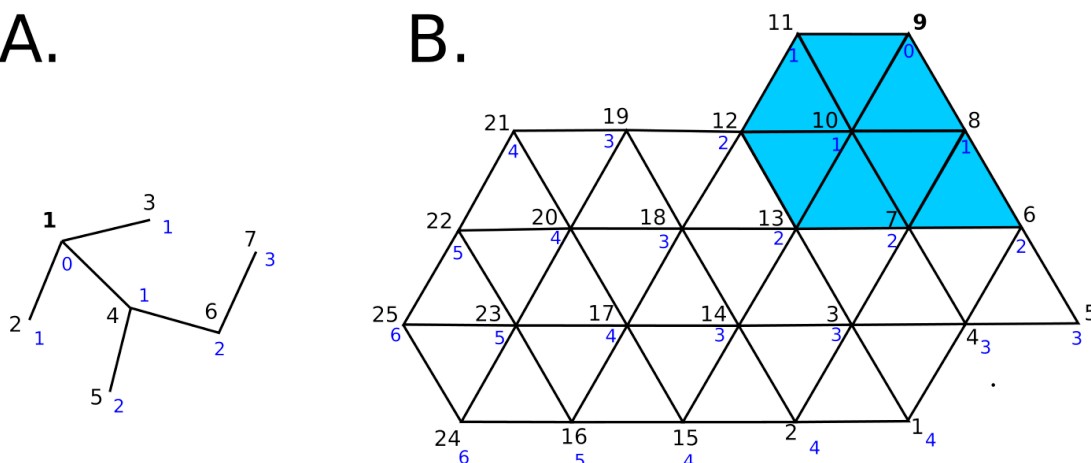

$$
\begin{array}{c|ccccccc}
\text{node} & 1 & 2 & 3 & 4 & 5 & 6 & 7 \\
\hline
1 & 0 & 1 & 1 & 1 & 2 & 2 & 3 \\
2 & & 0 & 2 & 2 & 3 & 3 & 4 \\
3 & & & 0 & 2 & 3 & 3 & 4 \\
4 & & & & 0 & 1 & 1 & 2 \\
5 & & \text{sym} & & & 0 & 2 & 3 \\
6 & & & & & & 0 & 1 \\
7 & & & & & & & 0 \\
\end{array}
$$

**Figure 5.** Distances between connected nodes in the graph-method. Node numbers are represented in black. (**A**) Distances from node 1 are shown in blue; (**B**) Distance from node 9 are shown in blue and triangles with nodes at a distance of less than two edges from node 9 are colored in cyan.

Distances from node 1 to all the other nodes correspond to the top line of the matrix and are indicated in blue in Figure 5, distances from node 2 to all other nodes correspond to the second line and so forth. A mesh of triangular elements like the one represented in Figure 5B is considered as a graph, from which the matrix distance can easily be computed with Matlab. Using this matrix, it is straightforward to pick nodes at a given edge distance from any specific node in order to select a subpart of the original mesh, which would constitute the pressurized source.

This method requires less parameters than the Projected Disk method as that part of the mesh which is submitted to pressure is fully described by only three parameters: a node number *n*, a distance *d* expressed as a number of edges and an overpressure *P*. The new mesh is a subpart of the original mesh centered on the selected node *n* and containing all the nodes located at a distance of less than *d*. For this method to lead to a circular pressurized area (the minimum level of a priori), triangular mesh elements need to be roughly equilateral.

## 5. Results

The static inversion results were evaluated based on the percentage of explained data (%*Ed*), which were computed per dataset or as an overall. For the *i*th dataset (either a given interferogram or the GNSS data), %*Ed^i* is evaluated as:

$$
\%Ed^i = \left( 1 - \sqrt{\frac{\left(\mathbf{u_{obs}}^i - \mathbf{u_{mod}}^i\right)^T \left(\mathbf{u_{obs}}^i - \mathbf{u_{mod}}^i\right)}{\mathbf{u_{obs}}^{i^T} \mathbf{u_{obs}}^i}} \right) * 100 \tag{9}
$$

where $\mathbf{u^i_{obs}}$ and $\mathbf{u^i_{mod}}$ are the data and model vectors corresponding to all pixels in the masked interferograms or to all GNSS measurements. This value is independent of the covariance matrix,

the subsampling and the dataset used to run the inversion. Based on this definition, the closer to 100%, the better the data fit.

### 5.1. Static Inversion

#### 5.1.1. Two Model Families Which Explain the Data Equally Well

Geometries of the best-fit model can be described by two families of models, which fit the data equally well (InSAR, continuous GNSS and campaign GNSS) for the whole eruption (Table 1). Because of the random search conducted, the best models belonging to one or the other family can result from inversions conducted with the same inputs as shown by Inv04a (F1) and Inv04b (F2). Family F1 corresponds to a curved fracture dipping to the east (blue model on Figure 6). Family F2 is a two-part intrusion consisting of a sill that turns into a dike before reaching the ground surface (red model on Figure 6). To better characterize these families, we also quantitatively compared geometries of the best-fit models by computing the average distance $D$ between a node and the closest node belonging to a second mesh.

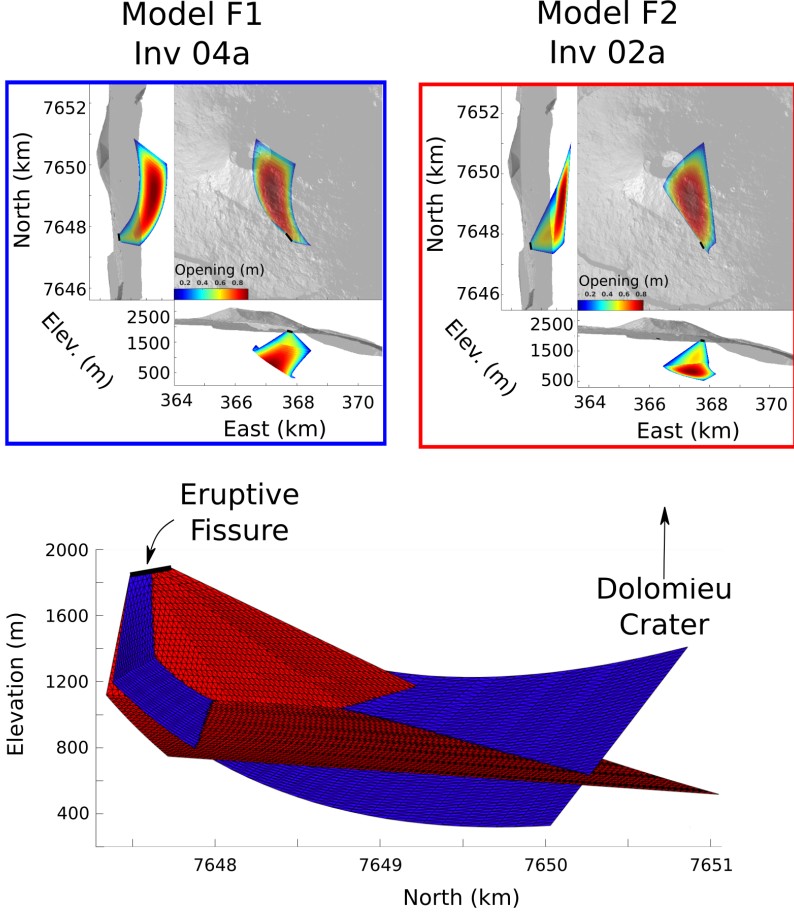

**Figure 6.** Comparison between geometries of typical intrusions of family F1 (Inv04a in blue) and family F2 (Inv02a in red). The upper plots compare the geometry and fracture opening of intrusions of family F1 (**left**) and family F2 (**right**). Three views are shown for each subfigure, one from the east along the northern direction on the left-hand side, a map view on the right-hand side and a view from the south along the eastern direction at the bottom. The lower plot compares the eruptive fracture meshes for the two families, as seen from the east.

Considering two meshes A and B with different node numbers, our average distance computation method (Figure 7) results in average distance from mesh $A$ to $B$, $D_{AB}$, being slightly different from the average distance from mesh $B$ to $A$, $D_{BA}$. Table 2 gives a summary of the average distance $D$ between pairs of meshes. From this table, we conclude that the average distance between meshes of family F1 is $94 \pm 32$ m, the average distance between meshes of family F2 is $103 \pm 45$ m and the average distance between families F1 and F2 is $160 \pm 35$ m. Meshes from a given family are closer to meshes from this family than those of the other family, indicating that our classification into families is relevant. The best model geometries for inversions 02b, 03 and 04b are very close and all belong to family F2. Model geometry of inversion 04a is close to 01 and 05. The three of them belong to family F1. The greater average distance between models 01 and 05 could be due to a systematic shift even though the overall shapes are very close. This shift is compatible with model 04a lying between model 01 and model 05 but closer to model 05. Model 02a also belongs to family F2. The closest model to model 02a is 02b, which also belongs to family F2 and the furthest from model 02a is 05, which belongs to family F1. Apparently large distances for models 03 and 04b are artifacts due to the averaging of small parts of the meshes which did not superpose well, whereas the central part of the mesh is in very good agreement.

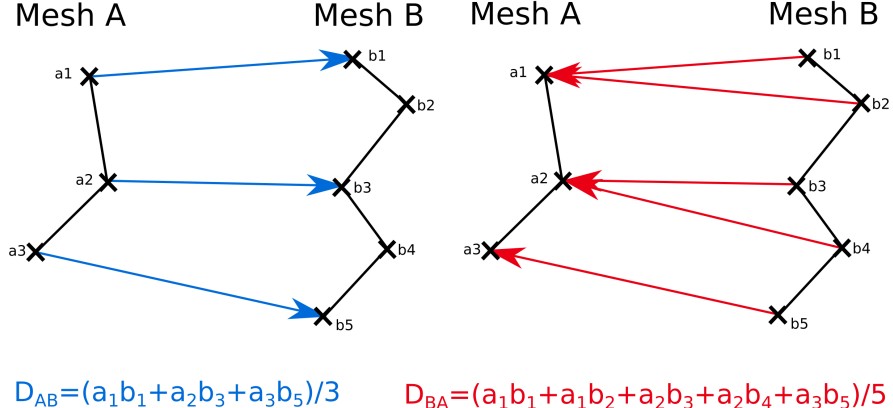

$$D_{AB}=(a_1b_1+a_2b_3+a_3b_5)/3 \qquad D_{BA}=(a_1b_1+a_1b_2+a_2b_3+a_2b_4+a_3b_5)/5$$

**Figure 7.** Methods for computing the average distance between two meshes, illustrating why $D_{AB}$ is slightly different from $D_{BA}$.

**Table 2.** Average distances (in meters) between pairs of meshes. * is the preferred model presented in Smittarello et al. [16]. Distances in bold are for models belonging to the same family.

| Inversion | | Inv01 | Inv02a | Inv02b | Inv03 | Inv04a | Inv04b | Inv05 |
|---|---|---|---|---|---|---|---|---|
| | Family | F1 | F2 | F2 | F2 | F1 | F2 | F1 |
| **Inv01** | F1 | X | 129 | 124 | 109 | **90** | 139 | **132** |
| **Inv02a \*** | F2 | 121 | X | **107** | **167** | 150 | **169** | 177 |
| **Inv2b** | F2 | 105 | **110** | X | **80** | 153 | **78** | 194 |
| **Inv03** | F2 | 104 | **153** | 72 | X | 157 | **51** | 202 |
| **Inv04a** | F1 | **85** | 141 | 168 | 167 | X | 185 | **63** |
| **Inv04b** | F2 | 116 | **144** | 61 | 46 | 164 | X | 209 |
| **Inv05** | F1 | **132** | 168 | 216 | 209 | **62** | 214 | X |

### 5.1.2. Importance of Consistency between Time Periods Covered

Inversion 01 was carried out using interferogram S1 A2, which does not cover the pre-eruptive period. The other inversions all used S1 A, which covers the same period as that spanned by the other interferograms. We find that S1 A consistently always gives explained data values between 8% and 12% higher than S1 A2 (Table 1). When inverting S1 A2, a posteriori computation of the explained data on S1 A, which was not used in the inversion, indicates a better fit than that of S1 A2. This clearly confirms that the data in S1 A2 are inconsistent with the other InSAR data. Given that S1 A2 does not cover the

pre-eruptive period, it is likely that the other interferograms, which do cover this pre-eruptive period (S1 D, S1 A, CSK D and CSK A) integrate a certain degree of pre-eruptive displacement.

### 5.1.3. Relative Weights of Ascending Versus Descending Interferograms

Without any weighting other than the data uncertainties (inversions 01, 02a, 02b, 03 and 05 in Table 1), the descending LOS always gave a better fit than the ascending LOS. The inversion algorithm searches for models that minimize the misfit and this minimization in the L2-sense favors a fit to the greatest displacement. There are two reasons for this (i) because our implementation of the quadtree subsampling does not account for the surface represented by the data, larger gradients and larger amplitudes of displacement lead to more points and thus have a higher weighting and (ii) because of the larger amplitudes at these points. This is demonstrated by the computation of the reference misfit with a null deformation model (Table 3). Considering only the InSAR data, the sum of the reference misfit on each interferogram leads to a total reference misfit of $\chi^{2i}_{ref} = 3070$, while the reference misfits for all ascending interferograms and all descending interferograms are 970 and 2100, respectively. Thus, although subsampled interferograms have a balanced number of points, the descending interferograms account for 2/3 of the total misfit. This higher weight simply results from the displacement pattern and amplitude and from the projection of this displacement along the LOS (Figure 8).

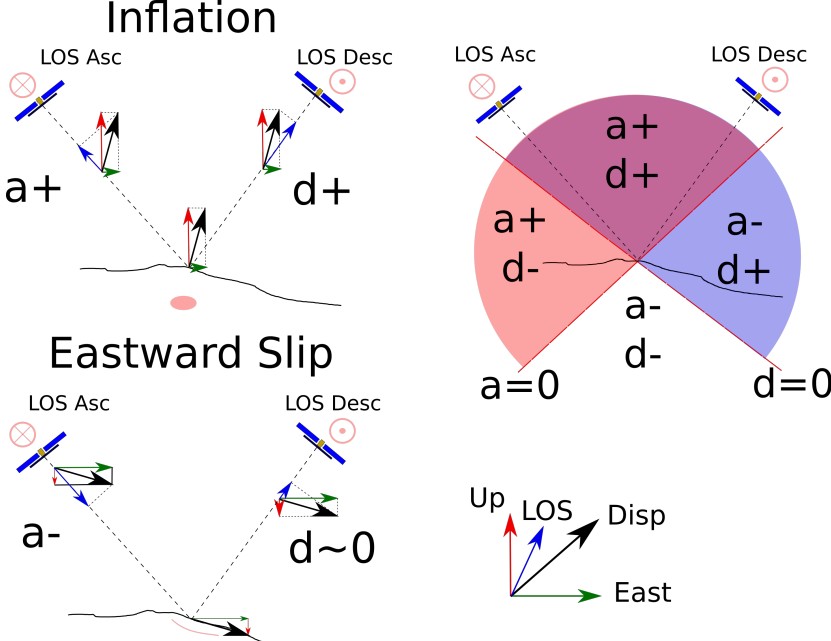

**Figure 8.** Projection of surface displacement along the LOS for satellite radar acquisition for ascending and descending orbits. The left-hand plots indicate expected displacements in the ascending and descending LOS for an inflation and an eastward flank slip. The right-hand plot shows the expected sign of the LOS projection (considered as positive when directed toward the satellite ) in the ascending and descending orbits for vectors joining a ground point to the circle.

To counteract the influence of the weighting, we normalized the data by their reference misfit $\chi^2{}_{ref}$. This weighting is equivalent to a normalization of the covariance matrix. Without this normalization, each dataset has uncertainties simply estimated from the data [57]. By altering this normalization factor, we used the values coming from the data uncertainties to evaluate the relative weighting of the data from a single dataset. However, we artificially considered that each dataset was equally likely. Results from inversions 03, 04a and 04b processed with this new weighting (Table 1) show an increase of the misfit on ascending data by a few percent, confirming that this weighting is efficient. Doing so, the calculated geometry remains the same and the inversion process is not able to improve the fit for the ascending data, even at the expense of the fit on the descending data.

**Table 3.** Values of $\chi^2_{ref}$ Equation (6) and weight of each dataset in the computation of the misfit. The top three lines correspond to inversions conducted without the campaign and continous GNSS data, while the bottom three lines include the GNSS data.

|  | GNSS | S1 D | S1 A | CSKD | CSKA | Total |
|---|---|---|---|---|---|---|
| $\chi^2 i_{ref}$ | X | 1100 | 500 | 1000 | 470 | 3070 |
| $\chi^2 i_{ref}/\chi^2_{ref}$ % | X | 36% | 16% | 33% | 15% | 100% |
| after weighting | X | 25% | 25% | 25% | 25% | 100% |
| $\chi^2 i_{ref}$ | 4800 | 1100 | 500 | 1000 | 470 | 7870 |
| $\chi^2 i_{ref}/\chi^2_{ref}$ % | 61% | 14% | 6% | 13% | 6% | 100% |
| after weighting | 20% | 20% | 20% | 20% | 20% | 100% |

### 5.1.4. Relative Weights of InSAR Versus GNSS Data

We also added campaign and continuous GNSS data to certain inversions. In inversion 05, the GNSS is weighted according to the measurements' standard deviations. Although GNSS data points are less numerous than InSAR data points ($\sim 80 + 10$ GNSS points with 3 displacement components for each point versus $\sim 2000$ InSAR points), they are independent of each other, while InSAR data are spatially correlated. Their variance on horizontal components was smaller ($\sigma_d^2 = 2 \times 10^{-4}$ m$^2$ and $\sigma_d^2 = 9 \times 10^{-3}$ m$^2$ on horizontal and vertical displacements, respectively) than that of InSAR ($\sigma_d^2 = 5 \times 10^{-4}$ m$^2$). This results in GNSS data accounting for 61% of the total reference misfit $\chi^2_{ref}$ Equation (6) and Table 3.

In inversions 04a and 04b, the covariance matrix was modified in order to give GNSS data the same weighting as each interferogram. This decreased the weighting of GNSS data to the total misfit from 61% in inversion 05 to 20% in inversions 04a and 04b (Table 3). However, the best model geometry remained the same regardless of the weighting and a similar fit of GNSS data was obtained (Table 1).

Model family F1 (inversion 04a) explains the GNSS data better (Table 1) than family F2 (inversion 04b). However this is done at the expenses of the InSAR fit giving the same total fit. Overall, all models explain the InSAR and GNSS data equally well and the addition of GNSS data does not provide any critical new information. With four different LOS, the displacement field is well characterized and any ambiguity is due to the non-uniqueness of inversions. Thus a criteria solely based on the misfit minimization is not sufficient to choose between the two equally well fitting families of models.

### 5.2. Temporal Inversion

#### 5.2.1. A Need for Geometrical a Priori to Invert for the GNSS Time Series

We developed three methods of source parametrization to invert the whole GNSS time series. Taking advantage of the SAR acquisition during the eruptive crisis, we compared inversion results using either continuous GNSS data, InSAR data (S1 D1) or both datasets covering the first part of the eruption (until 1:45) with each method (Table 4).

The Ellipse method does not have any a priori on the source location. The inversion of cumulated displacement from GNSS time series data leads to a very good fit (96%) of these data (Table 4). However, the fit on the S1 D1 interferogram falls to 22%. The horizontal location of the deformation source (Figure 9) is comparable to the source determined by the static inversion for the whole eruption. However, the source depth (Figure 9) and pressure (Table 4) are not accurately estimated because of a lack of GNSS stations close to the most deformed area. This leads to an overestimate of the pressure (Table 4) as well as the modeled displacement (Figure 10).

**Table 4.** Comparison of the inverted source characteristics for different methods for tracking magma propagation and different data: continuous GNSS data from the permanent network and an intermediate Sentinel-1 interferogram S1 D1 covering the first part of the eruption (before 01:45). % explained data (%*Ed*, see Equation (9)) computed with data used in the inversions are in bold, while those computed a posteriori using data omitted in the inversion are in small italics.

| Method Data | | Ellipse | | Projected Disk | | Subgraph | |
|---|---|---|---|---|---|---|---|
| | | GNSS | InSAR | GNSS | InSAR | GNSS | InSAR |
| Overpressure (MPa) | | 3.2 | 7.0 | 3.5 | 1.9 | 3.4 | 2.5 |
| Average opening (m) | | 1.3 | 1.3 | 1.0 | 0.6 | 1.0 | 0.7 |
| Area ($10^6$ m$^2$) | | 2.8 | 1.6 | 3.4 | 3.9 | 3.3 | 3.1 |
| Volume ($10^6$ m$^3$) | | 3.6 | 2.1 | 3.5 | 2.4 | 3.3 | 2.2 |
| %*Ed* | GNSS | **96** | *85* | **84** | *83* | **83** | *84* |
| | InSAR S1 D1 | *22* | **95** | *77* | **96** | *83* | **94** |

The Projected Disk and Subgraph methods, which rely on the a priori geometry determined in the static inversion step, give similar pressurized areas (Figure 9) and fit of InSAR data (Table 4 and Figure 10) when using the F2 model family. The fit for the GNSS data (Table 4) amounts to a high percentage of explained data of ∼84% and ∼83% for the Projected Disk and the Subgraph methods, respectively. With both methods, the modeled displacement fields are compatible with the S1 D1 interferogram but the percentage of explained data is slightly higher with the Subgraph (83%) than with the Projected Disk method (77%). With respect to the InSAR data, GNSS data overestimate the overpressure and the intruded volumes, which is related to the GNSS network configuration and coverage as indicated by the inversion of GNSS data always overestimating displacements (Figure 10) in the area which is devoid of GNSS stations. However, these overestimations are not as high with the Subgraph method (36% and 50% on the overpressure and volume, respectively) as with the Projected Disk method ((84% and 45% on the overpressure and volume, respectively), probably because the pressurized area is determined in a more precise way, free of any projection, with the Subgraph method.

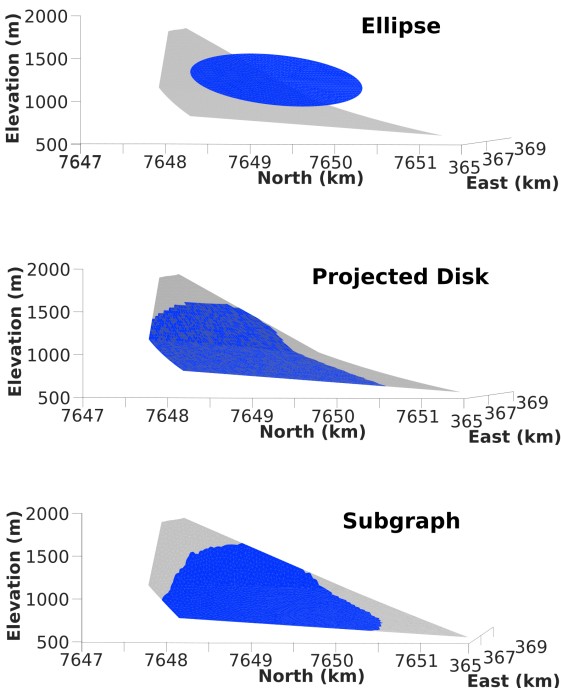

**Figure 9.** Comparison between the intrusion determined from InSAR data covering the whole eruption (grey) and the best model intrusion geometry (blue) resulting from the temporal inversion of GNSS data at 01:45. Results for the three different methods for inverting the pressure source are shown. (**Top**) Ellipse; (**Middle**) Projected Disk and (**bottom**) Subgraph methods.

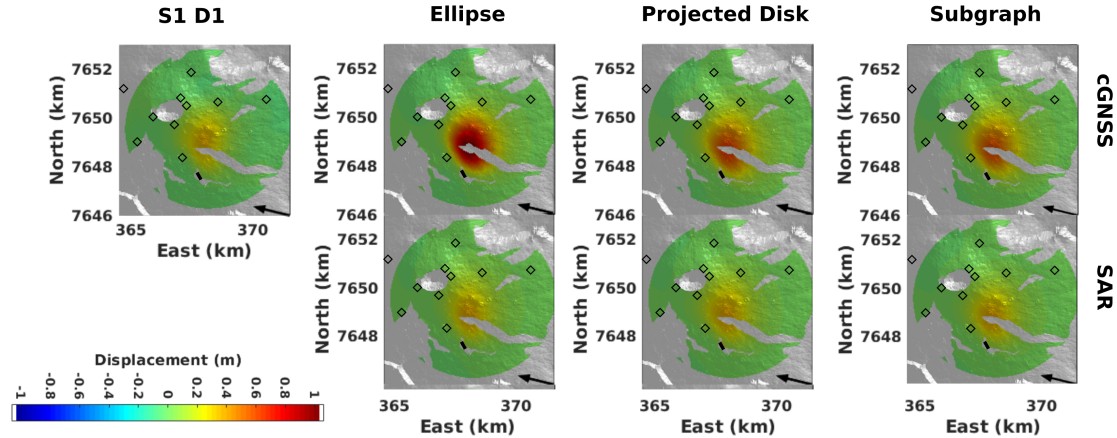

**Figure 10.** Comparison of inversion results for validation test using interferogram S1 D1. Wrapped interferogram S1 D1 (**first column**) and modeled interferograms computed with the best models determined with the Ellipse method (**2nd column**), Projected Disk method (**3rd column**) and Subgraph method (**4th column**). Inversion is done with continuous GNSS data alone (**top**) or with SAR S1 D1 data alone (**bottom**). Black segment markss the location of the eruptive fissure. Diamonds represent the location of permanent GNSS stations.

### 5.2.2. Inversion of GNSS Time Series to Improve Discrimination between Families of Intrusion Geometry

Two best-fit model families (F1 and F2) were determined with the static inversion: F1 characterized by curved sources dipping eastward and F2 characterized by a sill-to-dike geometry. We conducted the inversion of the GNSS time series with the Projected Disk method on geometries characteristic of F1

(inversion 04a) and F2 (inversion 02a). Inversions from F2 give a better fit than those of F1 (Figure 11) for all time steps with a larger difference at the beginning of the time series at 21:00 the misfits for F1 and F2 are $\chi^2 \sim 30$ and $\chi^2 \sim 20$, respectively and at 4:00 the misfits for F1 and F2 are $\chi^2 \sim 30$ and $\chi^2 \sim 25$, respectively. Pressures and openings determined for F1 are overestimated, reaching 10 MPa, while areas are underestimated, as indicated by the comparison with the result determined from the inversion of the intermediate interferogram (compare the blue curves with the open symbols at 1:45 in Figure 11). On the contrary, inversions of GNSS times series from F2 (Projected circle or Subgraph) show consistent results. Time series inversions also confirm that the Ellipse method leads to unreliable results in terms of pressure, area and average openings.

For both families, evolution of the source location with time (Figure 12) shows an increase in the source area and a migration from centrally beneath the Dolomieu crater to the southeast region of this crater. However, only the magma path determined from inversions with model F2 is consistent with a feeding system beneath the Dolomieu crater, followed by a lateral transport to the vent. With F1, magma travels back down, accumulates in the curved part of the source and never reaches the surface. The Subgraph method used with F2 has the greatest potential to give rise to an opening of the dike part of the eruptive fissure. The analysis of time series contributes critical information that helps resolve the non-uniqueness of deformation modeling. The F2 geometry is therefore considered to be the most realistic.

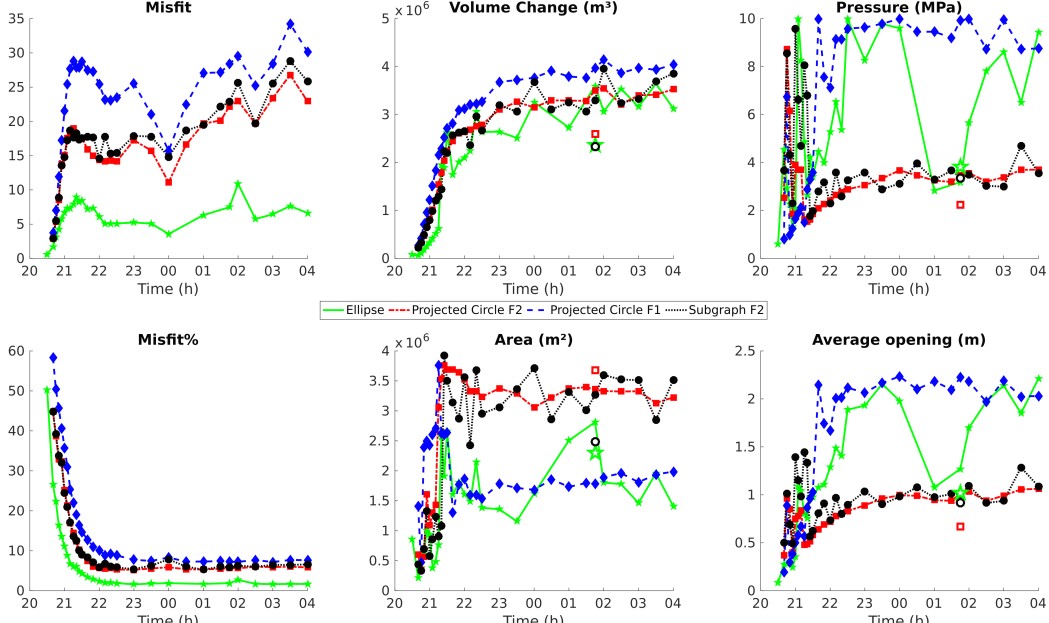

**Figure 11.** Results of inversions of the GNSS time series for 29 time steps between 20:40 and 4:00 a.m. Green line is the Ellipse method. Blue and red dashed lines are the Projected Disk method on a priori meshes for families F1 and F2, respectively. Solid black circles are the Subgraph method for family F2. Open symbols are for inversions from the intermediate Sentinel-1 interferogram S1 D1, green star corresponds to the Ellipse method, red square to the Projected Disk on family F2 and black circle to the Subgraph method on family F2. Note that for all methods, increase in the misfit between 20:40 and 21:30 is due to the increase in the amplitude of measured displacements as shown by the sharp decrease of the relative misfit in percent Equation (1).

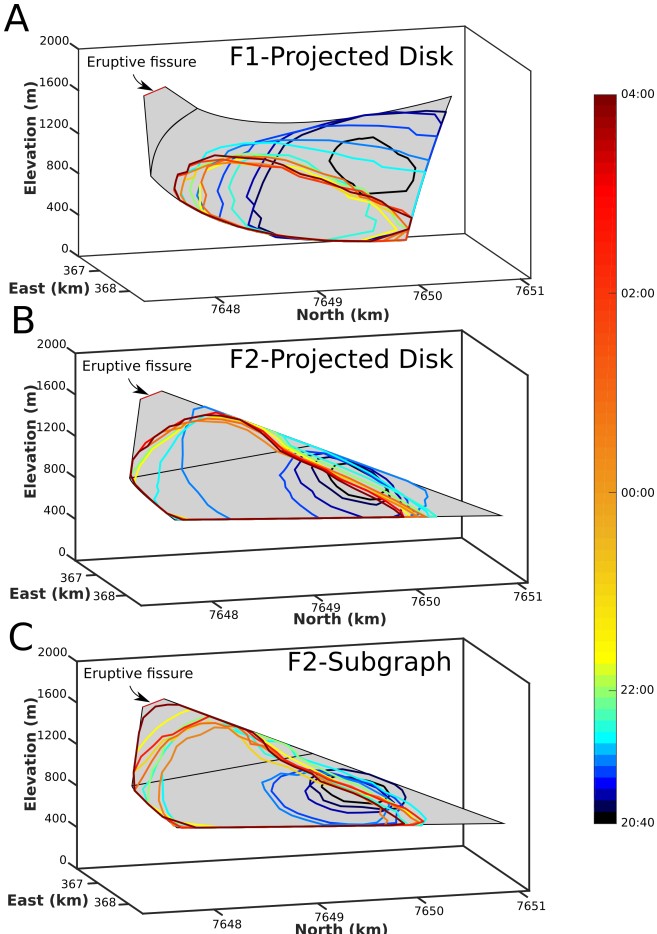

**Figure 12.** Location of the overpressure source determined from the temporal inversion (colors) for 14 time steps out of the 29 time steps inverted between 20:40 on 25 May and 04:00 the next day. The pressurized area is assumed to be a circular disk projected on the a priori mesh (in gray) of family F1 (**A**) and family F2; (**B**) or a subgraph of the a priori mesh of family F2; (**C**). Source geometries for F1 and F2 are non unique solutions determined from the inversion of the four sets of InSAR data that cover the whole eruption.

## 6. Discussion

### 6.1. Discrepancies between Independent Datasets Reveal Hidden Processes

#### 6.1.1. Discrepancies in the Covered Time Periods Reveal Pre-Eruptive Displacement

The recent increase in the frequency of SAR acquisition now makes it necessary to evaluate the selection of interferograms that span an eruptive event. Because of the difference in time scale and amplitude between pre-eruptive and co-eruptive deformation, the whole deformation pattern is generally assumed to reflect co-eruptive processes. An important question is whether it is more relevant to select interferograms that cover the same time periods or interferograms that are as close as possible to the eruption onset. In the present study, interferograms (CSK A, CSK D, S1 D and S1 A) that cover months before the eruption and captured the pre-eruptive period were available along with an interferogram (S1 A2) that only covered the few hours before the eruption. A systematically better fit on S1 A than S1 A2 reveals an inconsistency between S1 A2 and the other interferograms. During the 10 days preceding the eruption, a very slight inflation was noticed on the continuous summit GNSS stations (Figure 2). However, GNSS displacement being less than 1 cm, no fringes in interferogram S1 A1 can be conclusively associated with this inflation. Yet, the less good fit of S1 A2 than S1 A could come from the best model preferentially fitting the majority of the data, that is, the three interferograms

that recorded the pre-eruptive deformation and fitting the fourth interferogram, which which started after this deformation, less well. Therefore, the best model should be considered with caution as it might in reality represent a compromise between several displacement sources. Here we performed inversions with data from one time period and others incorporating data that covered a different time period to the rest of the data, showing that the latter approach can be an asset and may be used to highlight processes which are otherwise too subtle to appear clearly in the longer time-period data.

### 6.1.2. Discrepancies in Amplitude Along the Different Los Reveal Flank Displacement

We noticed systematic discrepancies between the fit of data acquired along ascending and descending LOS, with data acquired in the descending LOS being systematically better fitted than data acquired along the ascending LOS. Increasing the weighting of the ascending data did not produce different models or ones with a significantly better fit on the ascending LOS. We propose that this might be due to the ascending interferograms being more sensitive to a different source than the descending ones. For instance, a sill injection leads to ground inflation and produces a signal with similar amplitudes on the ascending and the descending LOS (Figure 8). However, an eastward movement of the eastern flank creates an eastward and downward directed displacement, which tends to cancel out on the descending LOS as opposed to cumulating on the ascending LOS. Such flank displacement could result from slip on a detachment fault [30] or a more complex host rock behavior for example due to plasticity [25].

### 6.2. Combining InSAR and GNSS for Complementary Spatial and Temporal Information

### 6.2.1. InSAR Static Inversion Constrains the Temporal Inversion

The Piton de la Fournaise permanent GNSS network is one of the densest GNSS networks in operation on a volcano. However, the case study of the May 2016 eruption reveals that using GNSS time series data to track magma may lead to an unreliable source pressure, area and opening when no a priori (here an elliptical source) are assumed about the deformation source. Indeed, displacement associated with this intrusion was only registered by 10 stations and the area where the displacement was the greatest was not monitored. In these conditions, the trade-off between the depth, pressure and size of the source cannot be resolved. InSAR is complementary to GNSS as it provides maps of displacement along the satellite LOS with a high resolution of 5 m. Combining displacement maps along several LOS allows a more complex magma intrusion geometry to be determined,which can then be used as a priori in the inversion of GNSS time series. We demonstrated that, with this a priori, we can better constrain the evolution of pressure, area and opening as shown by the comparison with the inversion of an interferogram covering the first part of the eruption.

### 6.2.2. Advantages of the Subgraph Method for Temporal Inversions

The Subgraph method is a new method developed to determine the pressurized part of a mesh from GNSS time series. In comparison with the Projected Disk method, which requires four parameters, only three parameters are needed, two for the geometry and one for the overpressure. As for the Projected Disk method, an a priori on the source geometry is needed, which is determined from the inversion of InSAR data covering the eruption. Results are very close to those obtained with the Projected Disk in terms of misfit and volume changes (Figure 11). However, comparing results with an intermediate Sentinel-1 interferogram, we find that the Subgraph method explains this data better than the Projected Disk method (Table 4). It also leads to a more realistic evolution of pressure, area and average opening. In terms of source geometry, the eruptive fissure never opens with the Projected Disk method (Figure 12). However, with the Subgraph method we are able to follow the propagation of magma upward to the surface. These results indicate that the Subgraph method, by avoiding the projection of a planar circle on a curved surface, is a more reliable method than the Projected Disk method.

### 6.2.3. GNSS Temporal Inversion Solves the Conundrum of Non Unique Static Inversion

In this study, we also showed that GNSS time series data help to choose between equally likely models determined from InSAR. Regardless of whether ascending and descending InSAR data have equal weighting or weighting coming from the data amplitudes or whether GNSS data were used or not, two families of equally well fitting models were obtained from the inversion of data covering the whole eruption (Table 1). These non unique results arise from inherent uncertainties pertaining to the data and the models. Here, we show that GNSS time series data can resolve the ambiguity between these families. In fact, the family of sources corresponding to a sill that turns into a dike leads to a source consistent with the displacement time series from GNSS and to a more meaningful propagation of the magma: magma starts beneath the summit and moves towards the ground surface in a way that is compatible with observations at this volcano [58]. The other family of sources corresponding to a curved eastward dipping intrusion is less consistent with the displacement time series and leads to magma going downward, accumulating at depth and never reaching the surface.

## 7. Conclusions

InSAR and GNSS data provide complementary information on the spatial distribution of displacements and their temporal evolution, respectively. Combining these data, we successfully inverted the source of observed displacements in order to track magma transport in the volcanic edifice of Piton de la Fournaise. We showed that four different LOS are sufficient to invert the static geometry of the emplaced magma intrusion. The addition of campaign and permanent GNSS displacement covering the same period does not remove the ambiguity pertaining to the non-uniqueness of inversions. Inversion of InSAR data covering a time period shorter to the other InSAR data indicates a pre-eruptive phase of deformation which is not detectible in the longer time period. Systematic discrepancies in data fitting between ascending and descending LOS are indicative of seaward flank displacement accompanying the magma intrusion. We also show that, in order to track magma propagation from GNSS time series, the intrusion geometry derived from InSAR measurements should be used as an a priori. A new graph-based method (the Subgraph method) is proposed to invert the pressurized part of the final geometry from GNSS time series. It provides a better fit for intermediate InSAR data than the two other methods: the Ellipse method which has no a priori and the Projected Disk method which uses the same final geometry as for the Subgraph method. It also leads to a more physically viable model where magma starts beneath the summit crater and propagates towards the surface, opening up eruptive fissures. When static inversions of InSAR data lead to equally well fitting models due to the inherent non-uniqueness of inversions, the GNSS temporal inversion may favor one family of models over the other. Systematic studies of new cases of propagating intrusions imaged by InSAR and GNSS times series would help increase our understanding of magma propagation. The shorter return time (of 6 days) for Sentinel–1 should make this possible in the coming years. Another possible approach for future studies would be to integrate the physics of magma propagation into a model before jointly inverting InSAR data and GNSS time series.

**Author Contributions:** Conceptualization, D.S., V.C. and V.P.; methodology, D.S.; software, V.C., D.S. and Q.D.; validation, V.C., V.P.; formal analysis, D.S.; investigation, D.S.; resources, V.P., V.C.; data curation, A.P., J.-L.F.; writing—original draft preparation, D.S., V.C. and V.P. ; writing—review and editing, V.C., V.P. and A.P.; visualization, D.S.; supervision, V.C., V.P. ; project administration, V.P., V.C.; funding acquisition, V.C., V.P., A.P. and J.-L.F.

**Funding:** The PhD Fellowship of D.S is funded by the French Ministry for Higher Education and Research. This research was partly supported by the INSU-CNRS ALEAS program, by the ANR through the SLIDEVOLC project (contract ANR-16-CE04-004-01) as well as the MagmaPropagator project (contract ANR-18-CE92-0037-01), by the French Government Laboratory of Excellence initiative ANR-10-LABX-00006, the CNES through the CNES project AssimSAR, the Région Auvergne and the European Regional Development Fund.

**Acknowledgments:** We thank the team which manage the OVPF, installs, maintains and monitors stations; F. van Wyk de Vries for proofreading the english of the paper. This paper has also benefited from discussion



with M. Acosta about the graph theory and A. Walsperdorf about GNSS data processing. Raw GNSS data are available on VOLOBSIS website (http://volobsis.ipgp.fr). Processed data are available by contacting the OVPF. We thank the European Space Agency for providing the Sentinel-1 data and the Italian Space Agency for providing the Cosmo SkyMed data. All interferograms used in this study were computed with Diapason ((c) TRE ALTAMIRA) by the Indian Ocean InSAR Observatory Service OI$^2$ , part of the National Service for Volcanological Observations (SNOV) of the French National Research Council Institute for Earth Sciences and Astronomy (CNRS/INSU). The interferograms are available on the web interface of the OI$^2$ platform by following CASOAR link (https://wwwobs.univ-bpclermont.fr/casoar). This is Laboratory of Excellence ClerVolc contribution number 375 and IPGP contribution number 4071.

**Conflicts of Interest:** The authors declare no conflict of interest. The funders had no role in the design of the study; in the collection, analyses, or interpretation of data; in the writing of the manuscript, or in the decision to publish the results.

## Abbreviations

The following abbreviations are used in this manuscript:

| | |
|---|---|
| ANR | Agence Nationale de la Recherche |
| GNSS | Global Navigation Satellite System |
| InSAR | Interferometric Synthetic Aperture Radar |
| LOS | Line of Sight |
| OVPF | Observatoire volcanologique du Piton de la Fournaise (Piton de la Fournaise Volcanological Observatory) |
| PdF | Piton de la Fournaise |
| PCA | Principal Component Analysis |
| SAR | Synthetic Aperture Radar |

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
