# Peer review of "Combining InSAR and GNSS to Track Magma Transport at Basaltic Volcanoes"

_remotesensing, doi:10.3390/rs11192236_

Round 1
Reviewer 1 Report
Author of manuscript: Smittarello et. al.
Manuscript ID: remotesensing-592240
Title of manuscript : Combining InSAR and GNSS to track magma transport at basaltic volcanoes
Comments and/or suggestions to the author:
The authors investigated the 2016 lava intrusion and eruption event at Piton de la Fournaise volcano with InSAR and GNSS data. I believe that readers of Remote Sensing have interest to this paper.
However, this paper is too long. If it possible, the shorter manuscript is better. For example, “the Ellipse method” description is not necessary.
The authors estimated that the magma propagation using GNSS time series. However, no validation was conducted of the estimation. Seismic data may help the validation.
The authors should revise their paper.
Author Response
We would like to thank reviewer 1 for the interest he had in this paper. We acknowledge that the paper is long but we do not believe this should be an issue and we consider that all paragraphs are necessary parts of our scientific demonstration.
In particular, the Ellipse method description consists in a very small paragraph (4lines) and the results are integrated with results provided by the other methos (Fiures 8 and 9) and shortly discussed. However, the use of this method is crucial to our demonstration that GNSS data only are generally too sparse on volcanoes to be inverted alone. Such inversion with no use of the a priori that could be brought by InSAR are a usual procedure (Fournier et al., 2009; Cannavo et al., 2015), so we believe this point is important. We added a sentence indicating that.
We agree that when available seismic data may help to constrain magma propagation. However, at Piton de la Fournaise, magma propagation appears not to be the primary source of seismicity (Duputel et al., 2018). Moreover, earthquakes are often too small and too shallow to be relocated precisely enough for this purpose. We added a phrase line XX to explain this and we refer to Smittarello et al, 2019 JGR where we showed the limitation of the seismic data for tracking magma before the May 2016 eruption of Piton de la Fournaise.
Reviewer 2 Report
This paper was a delight. I went through it several times and there is literally nothing that I can find wrong with it. I Never thought that would happen in my career, this is the first time in over 20 years.
After mulling it over though, I would suggest adding a reference or two from Zhong Lu. He literally wrote the book on InSAR, and has tried to use some ground based GPS to better constrain emplacement signals in InSAR, most notably on Alaskan volcanoes, Akutan comes to mind.
Otherwise well done!
Author Response
We are very grateful to reviewer 2 for his very positive review. We agree that adding reference to the work of Zhong Lu et al, is very relevant. We added references to the Chapter on the radar monitoring of volcanoes in the introduction at line 25 and to the determination of source depth from the joint inversion by Biggs, Lu et al., JGR 2010 at line 75.